# Study of Inducing Factors on Resveratrol and Antioxidant Content in Germinated Peanuts

**DOI:** 10.3390/molecules27175700

**Published:** 2022-09-04

**Authors:** Chun-Hsiang Hung, Su-Der Chen

**Affiliations:** Department of Food Science, National Ilan University, Number 1, Section 1, Shen-Lung Road, Yilan City 260007, Taiwan

**Keywords:** peanut, germination, resveratrol, antioxidant, rice koji, HPP, ultrasonic

## Abstract

When peanuts germinate, bioactive compounds such as resveratrol (RES), γ-aminobutyric acid (GABA), isoflavones, and polyphenol compounds are generated. Peanut kernels were germinated in the dark for two days, and stimuli including soaking liquid, rice koji, high-pressure processing (HPP), and ultrasonic treatment were tested for their ability to activate the defense mechanisms of peanut kernels, thus increasing their bioactive compound content. The results of this study indicate that no RES was detected in ungerminated peanuts, and only 5.58 μg/g of GABA was present, while unstimulated germinated peanuts contained 4.03 µg/g of RES and 258.83 μg/g of GABA. The RES content of the germinated peanuts increased to 13.64 μg/g after soaking in 0.2% phenylalanine solution, whereas a higher GABA content of 651.51 μg/g was observed after the peanuts were soaked in 0.2% glutamate. Soaking peanuts in 5% rice koji produced the highest RES and GABA contents (28.83 µg/g and 506.34 μg/g, respectively). Meanwhile, the RES and GABA contents of HPP-treated germinated peanuts (i.e., treated with HPP at 100 MPa for 10 min) increased to 7.66 μg/g and 497.09 μg/g, respectively, whereas those of ultrasonic-treated germinated peanuts (for 20 min) increased to 13.02 μg/g and 318.71 μg/g, respectively. After soaking peanuts in 0.5% rice koji, followed by HPP treatment at 100 MPa for 10 min, the RES and GABA contents of the germinated peanuts increased to 37.78 μg/g and 1196.98 μg/g, while the RES and GABA contents of the germinated peanuts treated with rice koji followed by ultrasonic treatment for 20 min increased to 46.53 μg/g and 974.52 μg/g, respectively. The flavonoid and polyphenol contents of the germinated peanuts also increased after exposure to various external stimuli, improving their DPPH free radical-scavenging ability and showing the good potential of germinated peanuts as functional products.

## 1. Introduction

*Arachis hypogaea* L. is the scientific name for peanuts (also known as groundnuts). In addition to their use as a food, peanuts can also be used to produce edible oil and peanut-series products. Peanuts contain approximately 40% fat, 25% protein, and 10% carbohydrates, along with lecithin, vitamins (i.e., A, B, E, and K), mineral elements (i.e., calcium, phosphorus, and iron), and eight types of essential amino acids for the human body [1]. In addition, peanuts contain phytonutrients with antioxidant and health-promoting properties, such as isoflavones, phenolic acids, phytosterols, and stilbenes. Peanut oil is composed of 15% saturated fatty acids, 50% unsaturated fatty acids, 30% polyunsaturated fatty acids, and 5% trans fatty acids; it is primarily composed of C18:1 oleic acid and C18:2 linoleic acid, which can effectively reduce cholesterol, retain good cholesterol, prevent heart disease, and eliminate harmful low-density lipoproteins [2].

During germination, peanuts grow, resulting in an increase in the weight of edible parts, a decrease in protein content, a slight initial increase in lipid content, and a significant initial decrease in the contents of soluble sugars and total phenols, followed by a significant increase. The contents of γ-aminobutyric acid (GABA) and resveratrol tend to increase with the germination time; consequently, germination treatment will enhance the nutritional value and biological activity of peanuts [3]. The germination ratio of peanuts will also be affected by the temperature of the soaking solution and the duration of soaking. The germination ratio is typically as high as 91.1% after 4 h of soaking and is the highest when the water temperature is maintained at 29 °C [4]. During peanut germination, the polyphenol and flavonoid contents and antioxidant activity of the hypocotyls are significantly higher than those of cotyledons. The highest levels of polyphenols and scavenging DPPH free radicals have been measured on the ninth day of germination, with the highest levels of flavonoids measured on the seventh day [5]. Germination can increase the antioxidant activity of peanut sprouts; however, the original kernel shape of germinated peanuts is easier to process and use. 

Resveratrol, a polyphenolic compound, is a naturally occurring stilbene phytoalexin that has protective effects against cardiovascular disease, aging, and cancer [6]. Sources of resveratrol currently include *Polygonum cuspidatum*, sprouted peanuts, and grapes. Resveratrol has the ability to scavenge DPPH free radicals and antioxidants, and has been shown in clinical trials to reduce oxidative stress, systolic blood pressure, and blood sugar levels, as well as having anti-inflammatory properties [7]. Resveratrol can inhibit amyloid-β-peptide [8], improve Alzheimer’s disease [9], and act as an anti-neurocytoma agent [10]. Furthermore, resveratrol-rich germinated peanut extract can alleviate benign prostatic hyperplasia and prostatitis [11]. During germination, peanuts not only produce resveratrol, but also decompose allergenic proteins, thereby reducing peanut allergy [12,13]. 

When plants are exposed to ultraviolet light, oxidative stress, fungal infection, and environmental stress, they produce resveratrol as a defense mechanism in order to resist invasion from the outside. The chemical structure of resveratrol is mainly divided into two structures: trans and cis forms. Resveratrol exists in its trans structure under natural conditions. Under irradiation with ultraviolet rays (UV at 254 nm or 366 nm), trans-resveratrol will convert into the cis structure; however, only trans-resveratrol is effective for human absorption. Therefore, peanuts must be germinated in the dark [14]. Resveratrol is produced through the synthesis of phenylalanine via phenylpropane [15].

Li [16] has infected peanuts with various fungi, including *Ganoderma lucidum*, koji, and *Rhizopus oryzae* in order to increase the resveratrol content. Notably, koji effectively increased the resveratrol content by 5–6 times. The phenol content also increased, which may be due to the rapid growth of rice koji, while the short fermentation time can increase the biotransformation activity of resveratrol glycosides during peanut germination, stimulating the production of resveratrol with potent antioxidant activity. Chen et al. [17] also mentioned that piceid-β-D-glucosidase from koji could rapidly convert albumin (polydatin) to resveratrol within 5 h. Moreover, after shaking the soaked peanuts for 20 min at 30 °C ultrasound, the germination ratio and total sugars slightly increased after three days of germination, while the crude lipids decreased, the protein content remained unchanged, and the resveratrol content increased. 

γ-aminobutyric acid (GABA) is an intermediate product of the catabolic pathway of glutamic acid. GABA is an important excitatory neurotransmitter in the central nervous system of animals; consequently, GABA can inhibit the transmission of central nervous substances and reduce pain or anxiety, promote sleep, and act as an antidepressant, among other functions. Furthermore, the pretreatment of soaked dried bean kernels (soy beans, black beans, red beans, and mung beans) with high-pressure processing (HPP) at 200 MPa for 10 min can increase the GABA content of sprouts [18]. Yang et al. [3] analyzed the changes of GABA content in five different peanut varieties during 5 days of germination. Due to the activation of decarboxylase—which converts glutamate to GABA during germination—GABA will continue to increase with germination time and its content may increase to more than eight times that in ungerminated peanuts.

The germination ratio is typically at its highest after 4 h of soaking. When the temperature of the soaking solution was maintained at 29 °C, the germination ratio reached as high as 90.9% [4]. Yu et al. [13] measured the content of resveratrol during peanut germination based on soaking time (2, 4, 6, 8, and 10 h) and germination time (1, 2, 3, 4, and 5 days). The resveratrol content changed over the germination period, and it was observed that the production of resveratrol continued to increase up to soaking for 6 h and germination for 1 day. 

The majority of the germination techniques involve soaking peanut kernels in liquid before germinating. There are four common soaking methods, whereby peanut kernels can be soaked in: (1) water; (2) inorganic salt solutions, such as NaCl, KNO_3_, CaCl_2_, or CaSO_4_ (3) solutions of sugars, sorbitol or mannitol, or polyethylene glycol (PEG); or (4) a pretreatment with various hormones such as salicylic acid (SA) or ascorbic acid (ASA), which affect the germination ratio. The germination status of peanuts soaked in SA and ASA was observed at 25 °C and 15 °C. At 15 °C, the germination ratios of peanuts soaked in SA and ASA were 97% and 98%, respectively, while the germination ratio of the control group was only 86% [19]. 

High-pressure processing (HPP) technology uses water as a medium for transmitting pressure at room temperature, with pressures of 100–1000 MPa, where the sealed packaged food is placed in an HPP environment. Physical pasteurization treatment is carried out at the proper time to destroy the non-covalent bonds, resulting in damage to the cell membrane structure of microorganisms, the formation of holes, and the subsequent loss of cytoplasm, which has a bactericidal effect, but also promotes germination. Changing the external pressure, temperature, and so on during germination will also affect the germination ratio. Peñas et al. [20] observed the germination ratio of mung bean seeds with varying temperature (10–40 °C) and pressure (100–400 MPa) for 10 min. The germination ratio of the mung bean seeds did not diminish when the temperature was in the range of 10–40 °C and the pressure increased to 250 MPa; however, when the pressure increased from 250 MPa to 400 MPa, the germination ratio of the mung bean seeds decreased significantly. 

The effects of different ultrasonic frequencies (i.e., 28 kHz, 45 kHz, 100 kHz) on the germination ratio of peanuts under three different peanut varieties (FH12, FH18, BS1016) indicate that the use of 100 kHz pretreatment had the strongest effect on peanut germination among the various varieties. The germination ratio of peanuts without ultrasonic pretreatment was increased by 5% compared to the control group; however, the ultrasonic pretreatment of peanuts improved their germination ratio and growth. Using various ultrasonic frequencies (28 kHz, 45 kHz, 100 kHz) in peanuts, the changes in the resveratrol content in germinated peanuts at different times (10, 20, and 30 min) for 1 to 5 days after germination were determined. The resveratrol content was the highest in all cases treated with ultrasonic vibration for 20 min during germination, and the resveratrol content was 31 μg/g on the third day under 45 kHz vibration for 20 min [13].

It is well known that peanut germination leads to the production of resveratrol, isoflavones, polyphenolic compounds, and γ-aminobutyric acid, and that their content increases with the germination time. In this study, we used various soaking solutions (including adding precursors and rice koji) and/or the use of HPP or ultrasonic pretreatment to stimulate peanut kernels in an attempt to produce higher resveratrol and antioxidant contents in the first 2 days of germination.

## 2. Results

### 2.1. Bioactive Substances in Peanuts during Germination

The appearances of the peanuts from 1 to 5 days after germination are shown in Figure 1, where the length of the bud from germination to the second day was about 0.5 cm, and the lengths of the third and fourth buds were 2.5 and 3 cm, respectively. By the fifth day, the length of the germ had increased to 5 cm and small fibrous roots had grown. Two days of germination is sufficient to consider that the germinated peanuts can be used for subsequent food processing in a prototypical manner.

As the moisture content of the germinated peanuts increased to 35% after soaking, it was necessary to air dry the germinated peanuts at 45 °C for 14 h to bring the moisture content close to 10% prior to storage and subsequent resveratrol and GABA content analysis (Figure 2). The resveratrol content in peanut is not detected internally, and is produced during germination. The peanuts contained 2.25 and 4.02 μg/g of resveratrol on the first and second days of germination, respectively, after which its generation slightly slowed down; however, the GABA content of the peanuts continued to rise from day one to day five of germination, from 258.8 μg/g on the second day of germination to 666 μg/g on the fifth day.

### 2.2. Effects of Different Soaking Solutions on the Bioactive Substances of Germinated Peanuts

After soaking Tainan 12 peanuts in water for 4 h and allowing them to germinate in the dark for 48 h, the germination ratio was 88% (Table 1). In ungerminated peanuts, no resveratrol was detected. However, the resveratrol and GABA contents of the dried germinated peanuts were significantly higher than those of the ungerminated peanuts (4.03 μg/g and 258.83 μg/g, respectively) after soaking in water for 4 h and germinating in the dark for 48 h. The germination ratio of the peanuts in various soaking solutions exceeded 85%. Sodium chloride solution (0.2%) stimulated peanut germination, increasing the contents of resveratrol and GABA to 5.51 μg/g and 363.79 μg/g, respectively. The 0.2% phenylalanine and 0.2% glutamic acid soaking solutions served as precursors to increase the biosynthesis of resveratrol and GABA during peanut germination, respectively. Consequently, soaking the peanuts in 0.2% phenylalanine significantly increased the resveratrol content to 13.64 μg/g, which was three times greater than that when soaking in water. Similarly, soaking peanuts in 0.2% glutamic acid also increased the resveratrol content to 8.70 μg/g, which was more than twice that when soaking in water, and the GABA content also reached 651.51 μg/g, which was approximately 2.5 times that when soaking in water. Moreover, after soaking peanuts in 5% rice koji for 4 h, the content of resveratrol and GABA significantly increased to 28.83 μg/g and 506.34 μg/g, respectively, due to the stimulatory effect. These values were the highest among all of the soaking solutions.

Table 2 shows that the contents of flavonoids and total polyphenols in ungerminated peanuts extracted with ethanol solution were 2.20 μg/g and 40.16 μg/g, respectively, and the ability to scavenge DPPH free radicals was only 26.90%. Germination contributed to the antioxidant capacity of the peanuts, as the contents of flavonoids and total phenolic compounds increased to 19.20 μg/g and 210.41 μg/g, respectively, while the DPPH free radical-scavenging ability increased to 42.80%, indicating that germination had a key contribution. With stimulation though the uses of four different soaking solutions to germinate the peanuts, 5% rice koji presented the highest flavonoid content (at 72.53 μg/g), followed by 0.2% glutamic acid (at 34.13 μg/g), 0.2% sodium chloride acid (at 31.23 μg/g), and 0.2% phenylalanine (at 27.82 μg/g). In terms of total polyphenols, 0.2% glutamic acid led to 1433.19 μg/g, 0.2% phenylalanine led to 993.35μg/g, and 0.2% sodium chloride had the lowest content (at 392.12 μg/g). In comparison to peanuts germinated in water, the antioxidant contents were significantly increased. In terms of scavenging DPPH free radicals, 5% rice koji had the highest ability (92.6%), followed by 0.2% glutamic acid (87.71%), 0.2% phenylalanine (79.29%), and the lowest was 0.2% sodium chloride (56.58%). It can be noted that the DPPH-scavenging ability with four different soaking solutions for peanuts was always above 50%, indicating that the ability to scavenge DPPH free radicals increased significantly compared to when soaking in water.

### 2.3. Effects of Different Physical Methods on Bioactive Substances in Germinated Peanuts

Table 3 provides the germination ratio and resveratrol content of peanuts germinated after being subjected to high-pressure processing (HPP). Using HPP to stimulate peanut germination, the germination ratio reached as high as 89% at 50 MPa, but gradually decreased as the pressure increased; notably, at 400 MPa, the germination ratio was only 49%. The content of resveratrol was the highest when the pressure was 100 MPa, reaching 7.66 μg/g, whereas a 400 MPa pretreatment only led to a resveratrol content of 5.34 μg/g.

Further using 50 and 100 MPa HPP with 5- and 10-min holding times under the same physical conditions stimulated peanut germination (Table 4). When the pressure was 50 MPa for 10 min, the germination ratio was the highest (89%). Under the conditions of 5 min at 50 MPa, 5 min at 100 MPa, and 10 min at 100 MPa, the germination ratio was 87%, and there was no significant difference between the three conditions. The resveratrol content of germinated peanuts treated at 50 MPa for 5 min was 5.47 μg/g, and the highest resveratrol content was 7.66 μg/g in the 100 MPa for 10 min HPP treatment, which was 1.9 times that of germinated peanuts not subjected to HPP. Therefore, appropriate HPP can enhance the production of resveratrol in germinated peanuts, facilitating resistance to the external environment.

The ungerminated peanuts only contained 5.58 μg/g GABA, while the germinated peanuts contained 258.83 μg/g GABA. The same phenomenon was observed as for resveratrol when HPP was applied. At 50 MPa for 5 min, the GABA content in the germinated peanuts was 354.15 μg/g, while at 100 MPa for 10 min, the GABA content was 497.09 μg/g.

Using ultrasound, the germination ratios of the peanuts were 89%, 88%, and 89% after 10, 20, and 30 min, respectively (Table 4). The resveratrol contents were 11.55, 13.02, and 8.67 μg/g under ultrasound after pretreatments of 10, 20, and 30 min, respectively. Increasing the duration of the ultrasonic vibration increased the GABA content proportionally. At 30 min of ultrasound, the highest GABA content was 357.89 μg/g, representing an increase of 1.38 relative to the untreated germinated peanuts. Therefore, using HPP and ultrasonic treatment was beneficial in terms of stimulating the resveratrol and GABA contents in the germinated peanuts. 

Table 5 shows that the contents of flavonoids and total polyphenols in the ungerminated peanuts were 2.20 and 40.16 μg/g, respectively, and the ability to scavenge DPPH free radicals was only 26.90%. After germination, the contents of flavonoids and total polyphenols in the peanuts increased to 19.20 μg/g and 210.41 μg/g, respectively, and their ability to scavenge DPPH free radicals also increased to 42.80%, indicating that germination is beneficial in terms of antioxidant activity. After peanut germination stimulated by HPP, the flavonoid and total polyphenol contents increased with pressure and the holding time of HPP (50 MPa for 5 min, 50 MPa for 10 min, 100 MPa for 5 min, and 100 MPa for 10 min). The flavonoid content increased significantly from 209.39 μg/g to 395.90 μg/g, and the total polyphenol content increased significantly from 883.13 μg/g to 1074.37 μg/g. Additionally, the ability to scavenge DPPH free radicals increased from 78.36% to 87.18%, indicating that the use of HPP is advantageous for enhancing the antioxidant activity of germinated peanuts.

In addition, the ultrasound stimulation of the germinated peanuts (10, 20, and 30 min; Table 5) slightly increased the flavonoid content from 20.65 μg/g to 30.89 μg/g, as well as the total polyphenol content from 244.28 μg/g to 303.79 μg/g. In terms of the DPPH free radical-scavenging ability, ultrasonic pretreatment for 20 min was the most effective (86.74%), followed by 10 and 30 min of stimulation (80.61% and 85.49%, respectively). Comparing the contents of flavonoids and total polyphenols after HPP and ultrasound, it is evident that the antioxidation activity-enhancing effect of HPP on the germinated peanuts was better, and the ability to scavenge DPPH free radicals was increased by about two-fold.

### 2.4. Effects of Physical and Chemical Methods on Bioactive Substances in Germinated Peanuts

Table 1 shows that soaking the peanuts in 5% rice koji (*Aspergillus oryzae*) solution significantly increased the resveratrol and GABA contents (by 7.15 and 1.96 times, respectively), when compared to soaking the germinated peanuts in water. When the peanuts soaked in water were subjected to 100 MPa for 10 min, the resveratrol and GABA contents increased by 1.9 and 1.92 times, respectively, when compared to the germinated peanuts soaked in water. The resveratrol and GABA contents of the germinated peanuts can increase by 3.23 and 1.23 times, respectively, when compared to soaking the germinated peanuts in water (Table 4). All of these conditioning stimuli increased the resveratrol and GABA contents in the germinated peanuts. The effects of the combination of physical stimuli and chemical soaking solutions on the resveratrol and GABA contents of the germinated peanuts are detailed in Table 6. The relevant method consisted of soaking the germinated peanuts in 5% rice koji liquid for 4 h, and then treating them with 100 MPa for 10 min or ultrasonic treatment for 20 min. The germination ratio reached as high as 89% after 48 h germination. The resveratrol content in the germinated peanuts increased to 37.78 and 46.53 μg/g, respectively, and the GABA content increased to 1196.98 and 974.52 μg/g, respectively, indicating that these combined methods could significantly increase the resveratrol and GABA contents of germinated peanuts.

In terms of antioxidant components, flavonoids, and total polyphenols (Table 7), germinated peanuts were soaked in 5% rice koji liquid for 6 h, and then treated with 100 MPa for 10 min or ultrasonically treated for 20 min, and continued to germinate for 48 h. The flavonoid content of germinated peanuts was 315.62 and 102.67 μg/g, and the total polyphenol content was 1992.89 and 1674.26 μg/g, respectively. Their ability to scavenge DPPH free radicals increased by about 89–90%, compared to 42.8% in the germinated peanuts soaked in water, indicating that the content of bioactive substances can be effectively increased by combining physical stimuli and chemical soaking solutions.

## 3. Discussion

### 3.1. Bioactive Substances in Peanuts during Germination

If it is considered that germinated peanuts can be used for food processing, it is best to germinate them for two days (Figure 1), rather than just considering the content of resveratrol and GABA (Figure 2). When analyzing the resveratrol content of germinated peanuts, it was indeed undetectable at the beginning of germination, but showed a linear increase from germination to the second day, with a production rate of approximately 2.01 μg/g day (R^2^ = 0.995, using regression). On day three, the resveratrol content increased to 4.89 μg/g, and then decreased slightly. In addition, the GABA content continued to increase from 1 to 5 days of germination, at a rate of 131.67 μg/g day (R^2^ = 0.982); notably, the GABA content on the second day of germination was 258 μg/g. For the follow-up study, two days of peanut germination were selected for assessment of changes in appearance and the rate of increase in resveratrol content during peanut germination.

Yu et al. [13] analyzed the effect of soaking time (2, 4, 6, 8, and 10 h) and germination time (1, 2, 3, 4, and 5 days) on the resveratrol content during peanut germination. It was observed that the amount of resveratrol produced in peanuts soaked for 6 h and germinated for 1 day remained relatively stable. Limmongkon et al. [6] analyzed five different varieties of peanuts (Kalasin 1, Kalasin 2, Konkaen, Konkaen 4, and Tainan 9), where the resveratrol content changed during 1–4 days of germination, and the resveratrol content trend also differed. Therefore, the highest resveratrol content, in terms of the germination day, differed for each variety of germinated peanut. The highest resveratrol content was 6.44 μg/g in Kalasin 2, which germinated for two days.

Yang et al. [3] also analyzed the changes in nutrients and bioactive components of five different peanut varieties throughout during the 5-day germination process. After two days of germination, the crude lipid content increased and the soluble sugar decreased, while the protein content remined relatively unchanged. However, on the fifth day of germination, the soluble sugar content increased, while the crude lipid content decreased significantly. As glutamate decarboxylase is activated, following which glutamate is converted into GABA during germination, the GABA content of the peanuts continues to increase during the fifth day of germination, eventually reaching more than eight times that in ungerminated peanuts. The resveratrol content increases with peanut germination time, but the ratio of increase varies between peanut varieties.

### 3.2. Effects of Different Soaking Solutions on the Bioactive Substances in Germinated Peanuts

No resveratrol was detected in the ungerminated peanuts and the GABA content was also trace. After soaking in water for 4 h, the peanuts were soaked in water for 48 h to germinate. The resveratrol and GABA contents in the cold-air-dried germinated peanuts were 4.03 μg/g and 258 μg/g, respectively. In order to increase the content of resveratrol and GABA, the peanuts were soaked in various soaking solutions, where 0.2% sodium chloride was the stimulus corresponding to a high salinity environment, while 0.2% phenylalanine and 0.2% glutamic acid are the precursors to resveratrol [15] and GABA [3,18], respectively, leading to higher amounts of resveratrol and GABA through biosynthesis. As for soaking peanuts in 5% rice koji, it is considered that microbial stimulation can induce higher amounts of resveratrol [16,17]; however, as there were only two days of germination in this study, the slower microbial growth rate may not be not appropriate.

The 0.2% sodium chloride soaking solution, as shown in Table 1, stimulated the germination of the peanuts, resulting in 1.4-fold increases in resveratrol and GABA contents compared to the germinated peanuts soaked in water. Yin et al. [21] stimulated the germination of soybeans by soaking them in salt water in order to observe the changes in GABA content. The GABA content of soybeans increased from approximately 300 μg/g to 410 μg/g after germination for 48 h. Due to the subsequent increase in the concentration of the external salt environment, the cells were forced to grow, and their GABA content increased. Zhu et al. [22] studied the effect of brine concentration on peanut germination and discovered that they are more sensitive to brine during the budding and seedling stages, while the salt tolerance concentration during the germination stage ranges between 0.2% and 0.3%. When the salt concentration exceeds 0.5%, the germination and growth of the peanuts will begin to be severely inhibited, and growth will almost halt in 1.0% salt water. As the salt concentration rises, the osmotic effect within the peanut also rises, causing the water required for growth to permeate the salt environment. As a result, the peanut’s ability to grow diminishes. Therefore, soaking peanuts in 0.2% salt water will have no effect on their growth and germination. Environmental stimulation can also increase the contents of resveratrol and GABA, improving the biofunction of germinated peanuts.

Soaking peanuts in 0.2% phenylalanine led to significantly increased resveratrol and GABA contents by 3.4 and 1.2 times, respectively, when compared to germinated peanuts soaked in water (Table 1). Meanwhile, the resveratrol and GABA contents of the germinated peanuts increased by 2.2 and 2.5 times, respectively, when soaked in 0.2% glutamic acid. Li [16] mentioned that phenylalanine is the precursor in resveratrol biosynthesis; therefore, soaking peanuts in phenylalanine solution can stimulate the peanuts to produce more resveratrol during germination. As resveratrol possesses anti-oxidation, anti-inflammatory, heart protective, anti-aging, and anti-cancer effects, the use of phenylalanine to obtain germinated peanuts for resveratrol enrichment provides a simple and convenient method for resveratrol accumulation.

Jiang et al. [23] reported that GABA is primarily produced by glutamic acid (Glu) though the catalysis of glutamate decarboxylase (GAD), with about five times higher GABA content than that in commonly germinated peanuts, demonstrating that soaking peanuts in glutamic acid can significantly increase GABA during germination. Adding the amino acid precursors of phenylalanine and glutamic acid in the soaking solution increased the contents of resveratrol and GABA in the peanuts through biotransformation during germination, as indicated by the results.

Due to its stimulatory effect, soaking peanuts in 5% rice koji for 4 h significantly increased the resveratrol and GABA contents, which were 7.2 and 2 times those of germinated peanuts soaked in water, respectively (Table 1). Li [16] used koji to infect peanuts to induce an elevation in the content of resveratrol; it increased the resveratrol content by 6.38 times, from 2.45 to 15.65 μg/g. The content of induced resveratrol was increased 7.18-fold, which may be attributable to the rapid concentration and growth of rice koji broth. Furthermore, the short fermentation time can improve the biotransformation activity of resveratrol during peanut germination, and the induced production leads to high levels of the antioxidant resveratrol. Uh et al. [24] also fermented germinated peanuts with lactobacillus for 15 h, which increased the resveratrol content in peanut drinks from 674 to 815 μg/L.

The content of flavonoids and total polyphenols extracted by ethanol in the peanuts after two days of germination were significantly increased (by 8.7 and 5.3 times, respectively), compared to peanuts without germination, and the ability to scavenge DPPH free radicals was also increased by 1.6 times (Table 2). Yang et al. [3] analyzed the changes in the total polyphenol content of five peanut varieties during five days of germination. They discovered that the content of phenolic compounds decreased slightly in the early stage of germination, but increased significantly from the second day of germination onwards. The total phenolic compounds content reached its maximum level on the fifth day of germination. Due to the fact that phenolic compounds can resist pathogenic bacteria in the growth and reproduction of plants, they can assist in maintaining the color of the material. Cevallos-Casals et al. [25] explained that germination occurs when the dry seed begins to absorb water, and as the embryonic axis grows, nutrients stored in the endosperm of the seed are supplied, in order to allow the seedling to grow, from the end of seed dormancy. The synthesis of factors can reveal the protective properties of phenolic substances and other compounds. Limmongkon et al. [6] utilized five different varieties of peanuts (Kalasin 1, Kalasin 2, Konkaen, Konkaen 4, and Tainan 9) to examine their polyphenol content and scavenging DPPH free radical ability between 1 and 4 days after germination. The total polyphenol contents were 30.81, 33.15, 22.49, 24.36, and 16.95 µg/g, respectively, and the scavenging DPPH free radical abilities were 57.93, 47.03, 53.79, 59.42, and 29.89%, respectively. Thus, there was a significant correlation between polyphenol content and scavenging DPPH activity.

Using four different soaking solutions to stimulate the germination of peanuts, the flavonoid content of the germinated peanuts soaked with 5% rice koji was the highest (4.3 times that with water), followed by 0.2% glutamic acid (1.8 times), 0.2% sodium chloride (1.6 times), and the lowest was 0.2% phenylalanine (1.4 times). In terms of total polyphenols, soaking 5% rice koji-germinated peanuts yielded the highest content (8 times that with water, followed by germinated peanuts soaked with 0.2% glutamic acid (6.8 times), 0.2% phenylalanine (4.7 times), and the lowest content was that with 0.2% sodium chloride; (1.9 times); however, all antioxidant contents were significantly higher than that of general germinated peanuts. The 5% rice koji led to the highest ability to scavenge DPPH free radicals (at 92.6%), followed by 0.2% glutamic acid (at 87.71%), 0.2% phenylalanine acid (at 79.29%), and 0.2% sodium chloride (at 56.58%), indicating that the ability to scavenge DPPH free radicals is significantly increased when compared to germinated peanuts soaked in water alone.

### 3.3. Effects of Different Physical Methods on Bioactive Substances in Germinated Peanuts

Table 3 demonstrates that peanuts pretreated with HPP for 10 min at 50 and 100 MPa did not affect the germination of peanuts, as both germination ratios were 87%. However, when the pressure was increased to 200, 300, or 400 MPa for 10 min, the germination ratio gradually decreased to 63–49%. The content of resveratrol was greater than that of the germinated peanuts without HPP treatment; however, increasing the pressure did not produce positive results. Appropriate pressure can have a significant stimulating effect, but excessive pressure will cause damage to the nutrients required for the internal growth of the peanuts. In comparison, Peñas et al. [20] used different HPP treatments (100–300 MPa) combined with hypochlorite and carvacrol to stimulate mung bean germination, where the seed germination ability decreased with increasing pressure under different concentrations. With concentrations of calcium chlorate and carvacrol of 18000 ppm and 1500 ppm, respectively, combined with an HPP of 250 MPa, the highest germination ratios were 80% and 60%, respectively. It was also confirmed that it is not the case that a higher pressure will lead to a higher growth ability and resveratrol content of peanuts. Consequently, it is necessary to select the optimal pressure to stimulate resveratrol production.

Under the conditions of holding the pressure at 50 MPa for 5 and 10 min, and holding the pressure at 100 MPa for 5 and 10 min, the holding time was increased from 5 min to 10 min regardless of the content of resveratrol and GABA (Table 4). With increasing pressure and holding time, 100 MPa and 10 min led to the highest resveratrol content of 7.66 μg/g and GABA content of 490.7 μg/g, both of which were approximately 1.9 times that of the germinated peanuts without HPP treatment.

For comparison, Chen et al. [18] soaked dried bean kernels (soybeans, black beans, red beans, and mung beans) at 200 MPa for 10 min, which increased the GABA content of the sprouts by 74.68, 93.29, 37.21, and 136.12 mg/100 g, respectively. Operating at 100 MPa can achieve effects similar to using 200 MPa for the same holding time. Therefore, using HPP to pretreat beans prior to germination changes the molecular structure of the proteins, thus affecting the contents of intracellular substances.

Ultrasonic treatment had little effect on the peanut germination ratio (remaining at about 88%); however, the resveratrol content was significantly higher than in the untreated germinated peanuts (Table 4). Ultrasonic treatment for 20 min produced the greatest amount of resveratrol: 3.2 times that in untreated germinated peanuts. The ultrasonic treatments produced peanuts with significantly higher resveratrol content than the HPP treatment. The GABA content also increased with ultrasonic treatment time, with the highest GABA content being 1.4 times that of untreated germinated peanuts after 30 min of ultrasonic treatment.

Comparatively, Yu et al. [13] used three different varieties of peanuts (FH12, FH18, and BS1016) with germination ratios of 90%, 92%, and 88%, respectively. During germination, the resveratrol content increased from 1 to 5 days. As for the changes observed in the three different peanut varieties treated with ultrasound for 20 min prior to germination, the contents of resveratrol were 15.06, 16.82, and 5.64 μg/g, respectively, on the second day of germination. Ultrasonication led to a rapid increase in the concentration of aniline lyase, which consequently increased the concentration of resveratrol in the germinated peanuts [26]. The increase in trans-resveratrol in peanuts exposed to ultrasound could be attributed to the induced plant cell defense response, which may explain the increase in resveratrol in germinated peanuts after ultrasonic treatment [27]. Overall, both HPP and ultrasound treatment increased the contents of resveratrol and GABA in the germinated peanuts, but HPP pressure was more conducive to increasing the content of GABA in germinated peanuts, whereas ultrasound increased the content of resveratrol in germinated peanuts.

HPP pretreatment significantly increased the flavonoid content in the germinated peanuts (Table 5); the flavonoid content of the peanuts treated with 100 MPa for 10 min was 20.6 times higher than that in the germinated peanuts without HPP. However, the flavonoid content in the germinated peanuts treated with ultrasound for 30 min was only 1.6 times than that in the untreated germinated peanuts; therefore, HPP was more effective at increasing the flavonoid content of the germinated peanuts. In addition, HPP pretreatment increased the total polyphenol content of the germinated peanuts by between 4.2 and 5.1 times that of the untreated peanuts, with the highest total polyphenol content occurring at 100 MPa for 10 min. The content of total polyphenols in ultrasonically pretreated germinated peanuts was only between 1.1 and 1.4 times that of the untreated germinated peanuts. The scavenging DPPH free radical ability of the ultrasonically pretreated germinated peanuts was increased to between 26.90% and 42.80% compared to the ungerminated peanuts. Using HPP pretreatment, the scavenging DPPH free radical ability of the germinated peanuts was increased to between 78.4% and 87.2%. Ultrasonic pretreatment could also greatly increase the scavenging DPPH free radical ability of germinated peanuts to between 80.61% and 86.74%. Thus, HPP and ultrasonic pretreatment can promote the production of antioxidants in germinated peanuts, thereby significantly enhancing their antioxidant activity.

In response to this phenomenon, Wang et al. [28] explained that the synergistic antioxidant response not only arises from the phenolic compounds, but also may come from other phytochemicals in the extract. In addition, the structure of the phenolic compounds has an effect on their antioxidant activity. The high correlation between total phenolic content and low molecular-weight compounds and the steric accessibility of antioxidant molecules to scavenge DPPH radicals represent important mechanisms for antioxidant evaluation.

### 3.4. Effects of Physical and Chemical Methods on Bioactive Substances in Germinated Peanuts

Yu et al. [3] discussed soaking peanuts in phenylalanine and ultrasonic pretreatment to increase resveratrol production during germination. Comparing soaking peanuts in 5% rice koji solution, soaking germinated peanuts in water, and using other soaking solutions (i.e., 0.2% brine, phenylalanine, or glutamic acid), soaking peanuts in 5% rice koji solution significantly increased the content of resveratrol and GABA (Table 1). When peanuts soaked in water were treated with HPP (100 MPa for 10 min) and the germinated peanuts were treated with ultrasound for 20 min, the content of resveratrol and GABA increased significantly compared to the germinated peanuts soaked in water alone (Table 4). In the combined chemical soaking solution and physical stimulation method, the peanuts were soaked in 5% rice koji solution for 4 h, and then treated with 100 MPa for 10 min or ultrasound for 20 min, and continued to germinate for 48 h. The contents of resveratrol and GABA were significantly increased, with the resveratrol content being 9.37 and 11.55 times that of the germinated peanuts soaked in water, respectively, and the GABA content being 4.63 and 3.77 times that of the germinated peanuts soaked in water, respectively (Table 6).

The flavonoid content in the germinated peanuts soaked in 5% rice koji liquid for 4 h, and then treated with 100 MPa for 10 min or ultrasonically treated for 20 min, and germinated for 48 h were 16.44 and 5.35 times that of the water-soaked germinated peanuts, respectively, while the total polyphenol content was 9.48 and 7.95 times that of the water-soaked germinated peanuts, respectively (Table 7). Their ability to scavenge DPPH free radicals was increased by approximately 89–90% from 42.8% for the germinated peanuts soaked in water, indicating that the contents of bioactive substances can be effectively increased by combining physical stimulation and a chemical soaking solution. In particular, the peanuts soaked in 5% rice koji, followed by ultrasonic treatment—which is simpler and less expensive than the HPP treatment of germinated peanuts—yielded good results. The results of this study have not been found in the previously published literature. This method is highly recommended in order to enhance the antioxidant properties of germinated peanuts.

## 4. Materials and Methods

### 4.1. Materials

The peanuts used in this experiment were purchased from Tainan No. 12, Zhuangwei Township, Yilan County. Sodium chloride (NaCl), L-Phenylalanine, L-Glutamic acid, 95% ethanol, methanol, sodium hydroxide (NaOH), gallic acid, and crystalline aluminum chloride (AlCl_3_·6H_2_O) were purchased from Wako Pure Chemical Industries, Ltd. (Osaka, Japan). Finally, 1,1-Diphenyl-2-trinitrophenylhydrazine, ascorbic acid, BHA (butylated hydroxy anisole), EDTA (ethylenediaminetetraacetic acid), and phosphomolybdic acid phenol reagent (Folin–Ciocalteu phenol reagent) were purchased from Sigma Chemical Company (St. Louis, MO, USA).

### 4.2. Equipment

We used cold-air-drying equipment (YK-112RS, Taiwan), an oven (Channel DCM45, Kehua Co., Ltd., Taiwan), ultra-high-pressure equipment (HPP600MPA/6.2 L, Kuntai International Co., Ltd., Taiwan), an ultrasonic oscillator (DC-600 H, DELTA, New Taipei City, Taiwan), an electronic precision scale (HDW-15 L, Hengxin Metrology Technology Co., Ltd., Taiwan), a high-speed grinder (RT-40, Kehua Co., Ltd., Taiwan), centrifuge (Z300, HERMLE, Germany), a high-performance liquid chromatography (HPLC, column specification: Ascentis C18, 25 cm × 4.6 mm × 5 µm), and a vacuum microwave extractor (Competitor Machinery Co., Ltd., Taiwan).

### 4.3. Preparation of Germinated Peanuts

A total of 100 g of peanut kernel seeds (with a water content of 10% or less), of commercially available dried Tainan No. 12 variety, were washed with tap water and soaked in tap water (control group), 0.2% phenylalanine, 0.2% glutamic acid, 0.2% salt, or 5% rice koji (*Aspergillus oryzae*) solutions, respectively, for 4 h. Then, the peanuts were spread out in a box and covered with a damp towel, taken out and rinsed once a day, and germinated in the dark. Germination was carried out in the dark for 2 days.

The induction of peanut kernel germination by HPP referred to the method of Chen et al. [18]. The 4-h-soaked peanut kernels were vacuum sealed and treated by HPP under 100, 200, 300, or 400 MPa for 10 min, respectively, at 25 °C. They were then germinated in the dark for 2 days.

The induction of peanut kernel germination by ultrasonic treatment referred to the method of Yu et al. [13]. In brief, the 4-h-soaked peanut kernels were sealed and treated by ultrasonic vibration in a tank for 10, 20, and 30 min, respectively, at 25 °C. They were then germinated in the dark for 2 days.

After completion, peanut budding was used as the criterion to calculate the peanut germination ratio (%), and the samples were dried at 45 °C in an air drier. The antioxidant substances were then analyzed.

### 4.4. Analysis of Antioxidant Components and Antioxidant Activity of Dried Germinated Peanuts

#### 4.4.1. Extraction and Analysis of Resveratrol

After drying, the germinated peanut kernel samples were pulverized and ground using a pulverizer. Then, 2 g of samples were weighed and placed in a glass test tube. After adding 10 mL of analytical-grade methanol, the samples were subjected to ultrasonic vibration extraction at 25 °C for 30 min, and centrifuged for 15 min (at 3870× *g*). After centrifugation, 1 mL of clarified solution was taken for filtration (0.2 μm) and placed in a glass sample test tube. With reference to the resveratrol analysis method [29,30], HPLC was used, with set detection conditions: the mobile phase is acetonitrile: water (35/65; *V/V*); flow rate, 1 mL/min; detection wavelength, 306 nm; column temperature, normal temperature; column specification, Ascentis C18 (25 cm × 4.6 mm × 5 μm). For HPLC, 0.5, 1.0, 2.0, 2.5, 4.0, and 5.0 ppm resveratrol standard solutions were prepared and analyzed, following which the extract samples were substituted into the standard curve to calculate the sample resveratrol concentrations.

#### 4.4.2. Extraction and Analysis of Gamma-Aminobutyric Acid (GABA)

The dried germinated peanut kernel samples were pulverized and ground with a pulverizer, and 2 g of the samples were weighed and placed in a glass test tube. After adding 10 mL of reverse-osmosis water for shock, the samples were extracted by ultrasound at 25 °C for 120 min, and then centrifuged for 15 min (at 3870× *g*). After centrifugation, 2 mL of the clarified solution was filtered (0.2 μm) and placed in a 2 mL plastic sample test tube.

Referring to the HPLC method [31] for the analysis of GABA, 2 mL of the clarified solution was filtered and 10 μL of derivatization reagent was added, and the reaction was performed for 2 min. The HPLC was set to the following detection conditions: the mobile phase as follows: Phase A—weigh 8.0 g of crystalline sodium acetate, quantify to 1000 mL with water, and then add 220 μL of triethylamine and stir. Use 5% acetic acid to adjust the pH to 7.2 ± 0.02, and finally add 5 mL of tetrahydrofuran. Phase B—weigh 8.0 g of crystalline sodium acetate, quantify to 1000 mL with water, and add 2% acetic acid to adjust the pH to 7.2 ± 0.02. Then, follow the ratio of sodium acetate solution: acetonitrile: methanol = 1:2:2, mixing by volume ratio; flow rate, 1 mL/min; detection wavelength, 338 nm; column temperature, 40 °C; column specification, Ascentis C18 (25 cm × 4.6 mm × 5 μm). Then, the gradient of the mobile phase was adjusted (as shown in the Table 8 below), and then standard solutions of 1.0, 2.5, 5.0, and 10.0 ppm of γ-aminobutyric acid were prepared for analysis. Finally, the extract samples were substituted into the standard curve in order to calculate the sample GABA concentration.

#### 4.4.3. Microwave Extraction

A total of 2.5 g of the germinated peanut kernel powder was added, 50 mL of 95% ethanol was added, and then 300 W microwave extraction was performed at a solid-to-liquid ratio of 1:20 for 5 min [32]. After centrifugation at 6000 rpm for 15 min, the supernatant was taken as the ethanol extract for the analysis of antioxidant components and activity.

#### 4.4.4. Determination of Total Polyphenols 

The total phenolic content of the peanut samples was determined by Folin–Ciocalteu’s assay [30]. To 1 mL of the extraction supernatant, 1 mL of Folin–Ciocalteu’s phenol reagent and 0.8 mL of 7.5% sodium carbonate were added, and then mixed well. The solution was then left to stand in the dark for 30 min at room temperature. The absorbance of the reacted mixture was observed at 765 nm using a UV-visible spectrophotometer. The total polyphenols in the peanut samples were expressed as μg gallic acid equivalents per gram of dried sample. 

#### 4.4.5. Determination of Flavonoids 

The flavonoid content in the peanut samples was determined using the aluminum chloride colorimetric method [33]. To 1 mL of the extraction supernatant, 1 mL of 2% methanolic (AlCl_3_·6H_2_O) was added and mixed well. The solution was then left to stand in the dark for 10 min at room temperature. The absorbance of the reacted mixture was measured at 430 nm, and the flavonoid content of the peanut samples is expressed as μg quercetin equivalents per gram of dried sample. 

#### 4.4.6. Determination of DPPH Free Radical Scavenging Activity 

The DPPH of the peanut samples was determined using a method modified from Xu and Chang [33]. To 2 mL of the extraction supernatant, 2 mL of 0.2 mM DPPH-MeOH solution was added and mixed well. Then, the solution was left to stand for 30 min under shading at room temperature. The absorbance was measured at a wavelength of 517 nm, which was substituted into the formula below to calculate the scavenging DPPH free radicals (%), where 5 mg/mL ascorbic acid, BHA, and EDTA were used as the control group. Calculation formula: scavenging DPPH free radicals (%) = ((ABS control − ABS sample)/(ABS control)) × 100%.

### 4.5. Statistical Analysis

The experimental data are expressed as mean and standard deviation. The data were analyzed using the Statistical Product and Service Solutions (SPSS, SPSS Inc., International Software Consulting Co., Ltd.) 20.0 statistical package software, including one-way analysis of variance (one-way ANOVA) analysis and multivariate full-range test analysis. (Duncan’s multiple-range test). The significance of differences in mean values was compared at the significance level of α = 0.05.

## 5. Conclusions

Germinated peanuts soaked in water alone for 4 h, and then germinated in the dark for 48 h, presented 4.03 µg/g of resveratrol and 258.8 μg/g of GABA. Using various soaking solutions, rice koji, HPP, and/or ultrasonic pretreatment to stimulate the germination of peanut kernels increased the resveratrol, GABA, flavonoid, and polyphenol contents to various extents, as well as the ability of scavenging DPPH free radicals. Overall, soaking peanuts in 0.5% rice koji, and then treating them with HPP 100 MPa for 10 min increased the resveratrol and GABA contents in the germinated peanuts to 37.78 μg/g and 1197 µg/g, respectively. In addition, in germinated peanuts after 4 h of rice koji solution soaking and then treatment with ultrasound for 20 min, the resveratrol and GABA contents increased to 46.53 µg/g and 974.5 µg/g, respectively. Taking into account cost considerations, it is suggested that peanuts be soaked in 0.5% rice koji solution for 4 h and then subjected to ultrasonic vibration for 20 min in order to significantly enrich their resveratrol and antioxidant contents. 

## Figures and Tables

**Figure 1 molecules-27-05700-f001:**
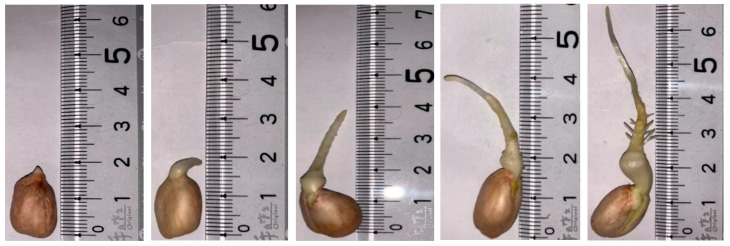
Growth of Tainan 12 peanuts germinated for 1 to 5 days.

**Figure 2 molecules-27-05700-f002:**
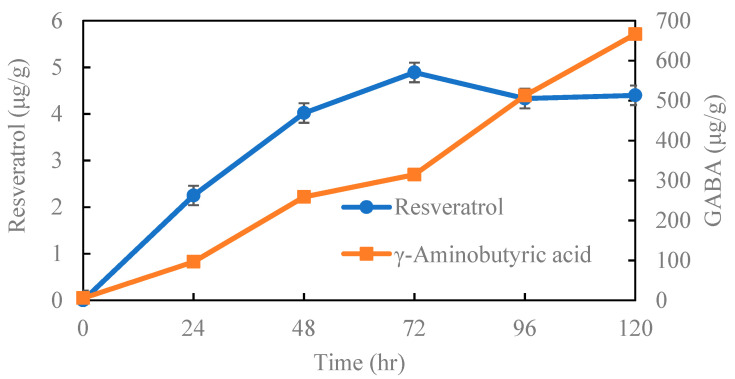
Changes of resveratrol and γ-Aminobutyric acid content in peanuts during germination.

**Table 1 molecules-27-05700-t001:** Effects of different soaking solutions on germination ratio, resveratrol, and γ-aminobutyric acid content of germinated peanuts.

Treatment	Germination Ratio (%)	Resveratrol (μg/g DM)	GABA (μg/g DM)
Ungerminated	-	-	5.58 ± 2.38 ^f^
Germination	88 ± 2.49 ^a^	4.03 ± 1.09 ^e^	258.83 ± 1.62 ^e^
Soaking solution			
0.2% NaCl	89 ± 2.02 ^a^	5.51 ± 2.35 ^d^	363.79 ± 2.43 ^c^
0.2% phenylalanine	91 ± 1.65 ^a^	13.64 ± 0.69 ^b^	312.75 ± 2.27 ^d^
0.2% glutamate	86 ± 0.75 ^b^	8.70 ± 1.31 ^c^	651.51 ± 1.46 ^a^
5% rice koji	87 ± 1.26 ^b^	28.83 ± 1.59 ^a^	506.34 ± 1.26 ^b^

1. Data are expressed as mean ± SD (n = 3). 2. ^a–f^ Means with different letters in the same column were significantly different (*p* < 0.05). 3. DW is dry material.

**Table 2 molecules-27-05700-t002:** Antioxidant active components and DPPH free radical-scavenging ability analysis of different soaking solutions on sprouted peanuts.

Treatment	Flavonoids (µg Quercetin Equivalent/g DW)	Total Polyphenols (µg Gallic Acid Equivalent/g DW)	Scavenging DPPH Free Radicals (%)
Ungerminated	2.20 ± 2.03 ^f^	40.16 ± 1.69 ^f^	26.90 ± 3.51 ^f^
Germination	19.20 ± 1.03 ^e^	210.41 ± 1.71 ^e^	42.80 ± 2.10 ^e^
Soaking solution			
0.2% NaCl	31.23 ± 1.54 ^c^	392.12 ± 1.28 ^d^	56.58 ± 1.12 ^d^
0.2% phenylalanine	27.82 ± 1.95 ^d^	993.35 ± 3.17 ^c^	79.29 ± 1.74 ^c^
0.2% glutamate	34.13 ± 0.81 ^b^	1433.19 ± 2.45 ^b^	81.71 ± 0.88 ^b^
5% rice koji	72.53 ± 4.3 ^a^	1694.22 ± 2.58 ^a^	92.60 ± 1.47 ^a^

1. Data are expressed as mean ± SD (n = 3). 2. ^a–f^ Means with different letters in the same column were significantly different (*p* < 0.05). 3. DW is dry material. 4. Concentration of DPPH sample was 10 mg/mL.

**Table 3 molecules-27-05700-t003:** Effects of high-pressure processing treatment under different pressures for 10 min on the germination ratio and resveratrol content of germinated peanuts.

Treatment	Germination Ratio (%)	Resveratrol (μg/g DW)
Ungerminated	-	-
Germination	88 ± 2.49 ^b^	4.03 ± 1.09 ^f^
High-pressure processing (HPP) holding time: 10 min		
50 MPa	89 ± 1.61 ^a^	6.67 ± 2.63 ^c^
100 MPa	87 ± 0.84 ^c^	7.66 ± 1.62 ^a^
200 MPa	63 ± 1.33 ^d^	6.91 ± 1.54 ^b^
300 MPa	57 ± 2.01 ^e^	6.22 ± 2.39 ^d^
400 MPa	49 ± 1.14 ^f^	5.34 ± 1.85 ^e^

1. Data are expressed as mean ± SD (n = 3). 2. ^a–f^ Means with different letters in the same column were significantly different (*p* < 0.05). 3. DW is dry material.

**Table 4 molecules-27-05700-t004:** Effects of high-pressure processing (HPP) and ultrasonic treatment on the germination ratio, resveratrol, and GABA content of germinated peanuts.

Treatment	Germination Ratio	Resveratrol (μg/g DM)	GABA (μg/g DM)
Ungerminated	-	-	5.58 ± 2.38 ^i^
Germination	88 ± 2.49 ^b^	4.03 ± 1.09 ^g^	258.83 ± 1.62 ^h^
High-pressure processing (HPP)			
50 MPa 5 min	87 ± 1.54 ^c^	5.47 ± 1.09 ^f^	354.15 ± 1.62 ^d^
100 MPa 5 min	87 ± 1.11 ^c^	6.62 ± 1.56 ^e^	397.42 ± 2.17 ^c^
50 MPa 10 min	89 ± 1.62 ^a^	6.67 ± 2.63 ^e^	467.95 ± 2.55 ^b^
100 MPa 10 min	87 ± 0.84 ^c^	7.66 ± 1.62 ^d^	497.09 ± 2.75 ^a^
Ultrasound			
10 min	89 ± 0.72 ^a^	11.55 ± 3.01 ^b^	273.40 ± 1.37 ^g^
20 min	88 ± 1.01 ^b^	13.02 ± 0.99 ^a^	318.71 ± 2.38 ^f^
30 min	89 ± 2.32 ^a^	8.67 ± 2.03 ^c^	357.89 ± 1.02 ^e^

1. Data are expressed as mean ± SD (n = 3). 2. ^a–i^ Means with different letters in the same column were significantly different (*p* < 0.05). 3. DW is dry material.

**Table 5 molecules-27-05700-t005:** Effects of HPP and ultrasound on the content of antioxidant active components in germinated peanuts.

Treatment	Flavonoids (µg Quercetin Equivalent/g DW)	Total Polyphenols (µg Gallic Acid Equivalent/g DW)	Scavenging DPPH Free Radicals (%)
Ungerminated	2.20 ± 2.03 ^h^	40.16 ± 1.69 ^i^	26.90 ± 3.51 ^i^
Germination	19.20 ± 1.03 ^g^	210.41 ± 1.71 ^h^	42.80 ± 2.1 ^h^
High-pressure processing (HPP)			
50 MPa 5 min	209.39 ± 1.53 ^c^	883.13 ± 2.33 ^d^	78.36 ± 1.64 ^g^
100 MPa 5 min	209.73 ± 2.01 ^c^	1024.49 ± 1.81 ^b^	81.54 ± 0.91 ^e^
50 MPa 10 min	229.69 ± 1.68 ^b^	1035.97 ± 0.99 ^c^	84.29 ± 0.72 ^d^
100 MPa 10 min	395.90 ± 1.33 ^a^	1074.37 ± 1.22 ^a^	87.18 ± 2.61 ^a^
Ultrasound			
10 min	20.65 ± 1.46 ^f^	244.28 ± 3.21 ^g^	80.61 ± 1.59 ^f^
20 min	23.72 ± 2.07 ^e^	281.57 ± 2.69 ^f^	86.74 ± 0.69 ^b^
30 min	30.89 ± 1.29 ^d^	303.79 ± 4.63 ^e^	85.49 ± 2.30 ^c^

1. Data are expressed as mean ± SD (n = 3). 2. ^a–i^ Means with different letters in the same column were significantly different (*p* < 0.05). 3. DW is dry material. 4. Concentration of DPPH sample was 10 mg/mL.

**Table 6 molecules-27-05700-t006:** Germination ratio, resveratrol, and GABA contents of germinated in rice koji with HPP or ultrasonic treatments.

Treatment	Germination Ratio (%)	Resveratrol (μg/g DM)	GABA (μg/g DM)
Ungerminated	-	-	5.58 ± 2.38 ^e^
Germination	88 ± 2.49 ^a^	4.03 ± 1.09 ^d^	258.83 ± 1.62 ^d^
5% rice koji	89 ± 0.63 ^a^	28.83 ± 1.59 ^c^	506.34 ± 1.26 ^c^
5% rice koji w/HPP	89 ± 0.41 ^a^	37.78 ± 2.11 ^b^	1196.98 ± 2.04 ^a^
5% rice koji w/ultrasound	89 ± 1.01 ^a^	46.53 ± 1.76 ^a^	974.52 ± 1.83 ^b^

1. Data are expressed as mean ± SD (n = 3) 2. ^a–e^ Means with different letters in the same column were significantly different (*p* < 0.05). 3. DW is dry material.

**Table 7 molecules-27-05700-t007:** Antioxidant components of germinated peanuts in with rice koji with HPP or ultrasonic treatments.

Treatment	Flavonoids (µg Quercetin Equivalent/g DW)	Total Polyphenols (µg Gallic Acid Equivalent/g DW)	Scavenging DPPH Free Radicals (%)
Ungerminated	2.20 ± 2.03 ^e^	40.16 ± 1.69 ^e^	26.90 ± 3.51 ^e^
Germination	19.20 ± 1.03 ^d^	210.41 ± 1.71 ^d^	42.80 ± 2.10 ^d^
5% rice koji	65.18 ± 1.33 ^c^	1511.47 ± 0.49 ^c^	86.61 ± 1.11 ^b^
5% rice koji w/HPP	315.62 ± 1.04 ^a^	1992.89 ± 0.98 ^a^	90.33 ± 1.75 ^a^
5% rice koji w/ultrasound	102.67 ± 1.85 ^b^	1674.26 ± 0.71 ^b^	89.14 ± 1.62 ^a^

1. Data are expressed as mean ± SD (n = 3). 2. ^a–e^ Means with different letters in the same column were significantly different (*p* < 0.05). 3. DW is dry material. 4. Concentration of DPPH sample was 10 mg/mL.

**Table 8 molecules-27-05700-t008:** HPLC mobile phase gradient table.

Time	Phase A	Phase B	Flow Rate (mL/min)
0	92	8	1.0
20	60	40	1.0
28	92	8	1.0

## Data Availability

Not applicable.

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
