# Peer review of "Study of Inducing Factors on Resveratrol and Antioxidant Content in Germinated Peanuts"

_molecules, 2022, doi:10.3390/molecules27175700_

Round 1
Reviewer 1 Report
This study mainly explored different methods (inlcuding using different soaking solution and phycial pretretment methods such as high pressure and ultrasound)to stimulate peanut kernels to produce higher resveratrol content after germination. Overall I think this study is interesting and valuable, but and language and writing still need to be improved. Several detailed comments:
1. Could you please explain what's difference between germination rate and germination ratio?
2. Line 43, GABA, first appearance here.
3. Line 92-94, rewrite this sentence.
4. Line 96-97, this sentence can be deleted since it has been mentioned in Line 46-48.
5. Line 100-102, rewrite this sentence.
6. Line 111, the germination ratio of the control group increased by 86%??? Please double check this statement.
7. Line 114, change 'sterilization' to 'pasteurization'.
8. Line 119-122, rewrite this sentence.
9. Overall the introduction part contains lots of information, but not well organized, please put more efforts to well organize this part.
10. Line 153-154, this sentence is very easy to misunderstanding, please rewrite it.
11. Line 151, 45℃ air drying is cold air drying?
12. Line 638, RES
13. In Conclution part, delete Line 641-647, and add one or two sentences to introduce other results instead of resveratrol and GABA.
Author Response
Thank the reviewer give us suggestion. We answer each problem and correct in the manuscript. Finally, the revised manuscript has undergone English language editing by MDPI. The revised manuscript with track changes was attached.
- Could you please explain what's difference between germination rate and germination ratio?
Ans: Germination rate is germination speed, and germination ratio is to present how many seed germination in %。
- Line 43, GABA, first appearance here.
Ans: γ-aminobutyric acid (GABA)
- Line 92-94, rewrite this sentence.
Ans: Due to the activation of decarboxylase, which converts glutamate to GABA during germination, GABA will continue to increase with germination time and may be more than eight times that of ungerminated peanuts.
- Line 96-97, this sentence can be deleted since it has been mentioned in Line 46-48.
Ans: We delete “The soaking solution and temperature will also influence the germination rate and antioxidant production of peanuts.”
- Line 100-102, rewrite this sentence.
Ans: The resveratrol content changed during peanut germination, and it was observed that the production of resveratrol continued to increase during soaking for 6 h and germination for 1 day.
- Line 111, the germination ratio of the control group increased by 86%??? Please double check this statement.
Ans: *****the germination ratios of peanut soaked in SA and ASA were 97% and 98%, respectively, but the germination ratio of the control group was only 86% [19].
- Line 114, change 'sterilization' to 'pasteurization'.
Ans: We change to pasteurization.
- Line 119-122, rewrite this sentence.
Ans: The sentence is changed to “The germination ratio of mung bean seeds did not diminish when the temperature was 10-40 °C and the pressure was increased to 250 MPa. However, when the pressure was increased from 250 MPa to 400 MPa, the germination ratio of mung bean seeds de-creased significantly.”
- Overall the introduction part contains lots of information, but not well organized, please put more efforts to well organize this part.
Ans: OK.
- Line 153-154, this sentence is very easy to misunderstanding, please rewrite it.
Ans: Resveratrol content in peanut was not detected at internal, and then produced during germination.
- Line 151, 45℃ air drying is cold air drying?
Ans: *** with air drying at 45 °C for 14 h****
- Line 638, RES
Ans: resveratrol
- In Conclusion part, delete Line 641-647, and add one or two sentences to introduce other results instead of resveratrol and GABA.
Ans: We delete it.
Reviewer 2 Report
I think that this work could be accepted for publication after minor revision. The detailed comments were shown as follows:
1. In general, improve the English language throughout the manuscript and some English revision is needed. There are some grammatical errors, particularly in respect of the use of tense and citation as well as instances of badly worded/constructed sentences. Some parts are confusing and inaccurate, particularly in the presentation throughout the manuscript.
2. The introduction was too long.
Author Response
Thank the reviewer give us suggestion, and we correct in the manuscript. The revised manuscript has undergone English language editing by MDPI. The revised manuscript with track changes was attached.